# Facilitators and Barriers in the Organ Donation Process: A Qualitative Study among Nurse Transplant Coordinators

**DOI:** 10.3390/ijerph17217996

**Published:** 2020-10-30

**Authors:** Víctor Fernández-Alonso, Domingo Palacios-Ceña, Celia Silva-Martín, Ana García-Pozo

**Affiliations:** 1Gregorio Marañón Sanitary Research Institute (IiSGM), International Doctoral School, Universidad Rey Juan Carlos, 28922 Madrid, Spain; victorferal@hotmail.com; 2Research Group of Humanities and Qualitative Research in Health Science of Universidad Rey Juan Carlos (Hum&QRinHS), Universidad Rey Juan Carlos, Alcorcón, 28922 Madrid, Spain; 3Gregorio Marañón Sanitary Reasearch Institute (IsSGM), 28007 Madrid, Spain; celia_1807@hotmail.com; 4General University Hospital Gregorio Marañón, Gregorio Marañón Sanitary Research Institute (IiSGM), 28007 Madrid, Spain; anagarciapozo8@gmail.com

**Keywords:** nursing care, tissue donor, organ transplantation, nursing, supervisory, qualitative research

## Abstract

Background: Spain is the world leader in organ donation, with a rate of 49.0 donations per million population. Nurse transplant coordinators fulfill key roles for the success of the complex donation process. Our aims were: (a) to describe the experience of nurse transplant coordinators and (b) to identify barriers and facilitators during the process of organ donation. Methods: A qualitative phenomenological study was conducted within the National Transplant Organization. A purposive sampling method was used, and data collection methods included semistructured interviews, researcher field notes, and participants’ personal letters. A systematic text condensation analysis was performed. The study was approved by the Clinical Research Ethics Committee. Results: A total of 16 participants were recruited and four themes were identified: (a) a different job for nurses, (b) facilitators and barriers of the coordinator’s job, (c) not a job for a novice nurse, and (d) coordinators facing a paradigm shift. Coordinators described their job as being characterized with uncertainty and having to face emotional and institutional barriers. The facilitators identified were high educational level and training, and feelings of pride for being part of the National Transplant Organization. Conclusions: The organ donation process requires specialized training to avoid organizational barriers.

## 1. Introduction

According to the Spanish National Transplant Organization (NTO), a “transplant” refers to the replacement of a diseased organ or tissue with another that works properly [1]. Without the solidarity of the donors, no transplants would be possible [1]. Spain is the world leader in organ donation, with a rate of 49.0 donations per million population (pmp), equaling 2302 organ donors who, together with living donors, enabled 5449 transplants, a rate of 116 pmp transplants [2]. The data from the latest report of 2019 by the NTO estimate that 68% of donations occur after brain death (DBD), whereas 32% of donations are a result of circulatory death (DCD). The donation rate in the Community of Madrid is below the national average, with a donation rate of 34.2 pmp. The rate of family refusals in Spain is 14%, equaling 15% in the Community of Madrid [2].

Organ transplantation is an exciting field that is subject to a large number of government regulations and public scrutiny [3]. The value of transplant coordinators is clearly recognized, fulfilling a robust and autonomous role as a key member of the multidisciplinary transplant team [3]. Transplant coordinators facilitate patient care throughout all phases of transplantation [3], ensuring ethical practice in transplant coordination and nursing [4]. In Spain, nurses have developed a significant role in the NTO with three major competencies: (a) nursing professionals value the possible donor, (b) obtaining consent, and (c) evaluating the presence of contraindications [1]. Spanish nurse transplant coordinators (NTCs) play a significant role in the NTO, particularly in relation to the coordination of organ donation and transplantation, data and information management, dissemination, training, and research [5]. According to the law, the duties of the NTO include procurement coordination and national distribution and international exchange of organs, tissues, and cells for transplantation [6]. The main duty of the hospital transplant coordination care unit is the organization and optimization of the collection and clinical use of human organs [6]. Likewise, the role of the NTC in the Regional Office for Transplant Coordination (ROTC) [1] is to:
“collaborate in the fulfillment of general objectives set by the Commission of Transplants of the Inter-territorial Council, and coordinate actions at the hospital level and at the level of the targeted population for the promotion of donation and transplantation of organs and tissues… they also assume the coordination of resources, information, dissemination, promotion, continuing education, as well as research and cooperation with other institutions”.(p. 42)

In this manner, the NTC plays a fundamental role in the identification, confirmation, and notification of potential transplants and increases the probability of the family’s consent to donate [7]. This results in the net system of the Spanish regional, national, and hospital coordination model improving the donation rate [8]. The presence of the transplant coordinator in ICUs is defined as a facilitator or key role for the success of the organ donation process [9]. Previous studies have analyzed the experience of transplant coordinators. Thus, transplant coordinators show a higher compassion fatigue/traumatic stress score compared with normal healthcare staff [10]. Indeed, suffering from compassion fatigue/traumatic stress influences patient attention [11]. The emotional consequences of this role constitute barriers that can increase burnout and can seriously affect both retention rates and the organ donation rate [12]. Conversely, transplant coordinators were more satisfied than transplant staff nurses regarding job independence and freedom regarding the physical demands of the job [13].

In Spain, Danet Danet et al. [14,15] described the transplant coordinator’s work experience as a “double perception” between satisfaction/pride and stress/pressure. In addition, certain aspects facilitate the work of the coordinator such as: training, professional quality, and interprofessional relationships in the coordination team, together with the success of the transplant results. However, emotional barriers are also described, such as constant work with death and grieving, and institutional barriers, such as organizational and relational difficulties at the professional level [14,15]. Our study gathers the exclusive experience of nursing professionals working as transplant coordinators. In addition, this study seeks to understand the global experience of the donation process, which is why participants were included from hospital transplant coordination units, as well as regional and national coordinators in Madrid that perform out-of-hospital coordination activity. Unlike previous studies such as the paper by Danet Danet et al. [14,15], the purpose of the present study was to describe the experience of NTC in the process of organ donation. Additionally, this study sought to analyze the current barriers and facilitators faced by NTCs in the process of organ donation.

## 2. Materials and Methods

The guidelines for conducting qualitative studies established by the Standards for Reporting Qualitative Research (SRQR) were followed [16]. Qualitative methods are useful for understanding the beliefs, values, and motivations that underlie individual health behaviors [17].

### 2.1. Design

A qualitative phenomenological descriptive design was conducted following Husserl’s framework [17]. In the field of qualitative studies, phenomenology attempts to understand how individuals construct their world view. Thus, it looks through a window into other people’s experiences. Phenomenology attempts to identify the essence of participants’ lived experiences, which is the subjective reflection on human beings when taking part in events in a specific space and time [18]. Husserl’s framework [19,20] guided this study. For Husserl, the aim of phenomenology is the study of aspects as they appear to reach an essential understanding of human experience [19,20]. To consider subjective experiences, the researcher assumes a certain attitude of attentive openness and readiness for a proper understanding of the unique meaning of participants’ lived experiences [21]. This experience always has a meaning for the person who has undergone the same, and thus, phenomenological research uses first-person narratives from the participants themselves as a data source [17,21].

### 2.2. Research Team

Four researchers participated in this study, three of whom had experience in qualitative study designs (C.S.-M., D.P.-C., A.G.-P.). Prior to the study, the researchers’ positioning was established via two briefing sessions addressing the theoretical framework for the study, their beliefs, and their motivation for the research [17,21]. Researchers based their approach on an interpretivist paradigm. This paradigm was based on the assumption that human beings construct their own social reality, and that knowledge is built through increasingly nuanced reconstructions of individual experiences. Also, researchers believed that the NTC plays a significant role in the Spanish model for donation and transplantation. The care of the deceased donor is the key to the success of the donation process. Finally, researchers believed that nursing care during the organ donation process would be able to define improvements and advances that positively influence the increase in the donor pool.

### 2.3. Context

The Spanish public health system centralizes and directs organ donation. This requires coordination between viable donors, the family, and the donation process. This donation process can be subject to stressors and barriers that delay donation. In this context, the role of the NTC is fundamental to eliminate barriers and empower facilitators that allow donation. However, the NTC can be subject to professional difficulties in public hospitals, such as the lack of recognition, or the lack of its own space within the donation process.

### 2.4. Participants

The inclusion criteria consisted of nurses who have worked in transplant coordination for at least one year. Nurses with less than one year of work experience in transplant coordination and those with serious psychiatric or cognitive disorders, or inability to communicate in Spanish or provide informed consent were excluded, as well as those who did not wish to participate in the study.

Purposive sampling was used, based on relevance to the research question (not clinical representativeness) [17,21]. Purposive sampling can be defined as the selection of individuals based on specific purposes associated with addressing the research study’s question or aim [17,21]. Sampling and data collection were pursued until the researchers achieved information redundancy, at which point no new information emerged from the data analysis [17]. In our study, this situation occurred after including 16 participants.

The researchers were introduced to the NTCs through the director of the nursing department at the NTO, the ROTC, and the six public Transplant Hospitals of Madrid. Previously, the director of the nursing department explained the purpose and design of the study to the NTCs who met the inclusion criteria. A one-week period was then allowed for patients to decide whether they wished to participate. In a second face-to-face session, the NTCs were asked to provide written informed consent and permission to tape the interviews. All the selected NTCs agreed to participate in the study. There were no dropouts. A total of 16 NTCs were recruited: six from NTO, one from ROTC, and nine from Madrid transplant hospitals.

### 2.5. Data Collection

Data were collected over a four-month period between February and April 2019. Data collection consisted of semistructured interviews based on a question guide (Table 1) in order to obtain information regarding specific topics of interest [16]. The question guide was developed based on a literature review and previous experiences of researchers. Subsequently, the researchers listened carefully, noting the key words and topics identified in the participants’ responses and using their answers to ask further questions for clarification. In this manner, relevant information was collected from the participants’ perspective [21]. Additionally, the researchers used prompts: (a) to encourage the participants to go into further detail (“Can you expand more on that?”), (b) to encourage the participants to continue talking (“Have you experienced the same thing since?”), and (c) to dissipate confusion (paraphrasing something that the patient had said) [21]. The interviews were conducted by V.F.-A. The interviews were audio-recorded and transcribed verbatim. A total of 16 interviews were undertaken (one per nurse). All interviews were conducted at the nurses’ workplace, according to participant preference. A reflexive journal kept by the researcher was also collected. The researcher’s field notes provided a rich source of information, by describing participants’ personal experiences and their behavior during data collection, and enabling the researcher to note any reflections concerning methodological aspects of the data collection [17,21]. During data collection, the participants were interviewed on their own.

### 2.6. Data Analysis

Complete verbatim transcripts were produced for each of the interviews, and researcher field notes. The texts were collated to enable qualitative analysis [17,21]. The initial analysis was conducted by CSM, AGP, and VFA. The initial results were subsequently merged in joint sessions, during which data collection and analysis procedures were discussed. In the case of differences of opinion, theme identification was decided by consensus. A systematic text condensation analysis was performed [22]. Systematic text condensation (STC) is an elaboration of Giorgi’s principles (a follower of Husserl) [18], including four comparable steps of analysis [22]. In this descriptive approach, presenting the experience of the participants as expressed by themselves, and following Giorgi, STC provides an explorative proposal to present vital examples from peoples’ life worlds [22]. Also, like Giorgi’s method [19,20,23,24], STC implies analytic reduction with decontextualization and recontextualization of data [22]. This procedure consists of the following steps [22]: (1) reading all the material and an overview of the data; (2) reviewing the transcript line by line to identify meaning units. A meaning unit is a text fragment containing some information about the research question. Subsequently, coding begins (decontextualization), which includes identifying, classifying, and sorting meaning units and marking these with a code—a label that connects related meaning units into a code group; (3) implying the systematic abstraction of meaning units within each of the code groups established in the second step of analysis. Empirical data are reduced to a decontextualized selection of meaning units sorted as thematic code groups across individual participants; (4) data are reconceptualized, putting the pieces together again. Themes are developed, providing stories that reflect the participants’ experiences [22]).

The process began with the most descriptive content in order to obtain meaningful units. Subsequently, a more in-depth analysis took place by using data reduction in order to classify these into thematic code groups (i.e., grouping of meaningful units referring to the same point or content until the main topics emerged). In this manner, the level of abstraction and complexity of the analysis increased from meaning units to thematic code groups, and finally themes [22]. An example of this process is described in Table 2. The final outcome was the identification of themes that represented the NTC’s experiences. No qualitative software was used on the data.

### 2.7. Rigor

The guidelines established by the SRQR were followed [16], together with the criteria for guaranteeing trustworthiness as cited by Guba & [17,21]. The techniques performed and the application procedures [17,21] used to control trustworthiness are described in Table 3.

### 2.8. Ethics

The study was approved by the Clinical Research Ethics Committee of Hospital Universitario Gregorio Marañón (code: VFA_ENF; act: 02/2019). Prior to the study, informed consent and permission to record the interviews were obtained from each participant. Furthermore, this study was conducted in accordance with the principles of the Helsinki Declaration.

## 3. Results

Twelve females and four males were enrolled in this study. The mean age of participants was 50.44 years. The mean years of experience as nurses was 27 years (SD 8.13) and the mean years working in transplant coordination was 10.5 years (SD 7.25). Participants’ narratives were taken directly from the interviews regarding the four emerging themes. Table 4 reports themes identified from participants’ narratives.

### 3.1. A Different Job for Nurses

The NTC’s job is perceived as different to the role of healthcare nurses. However, this difference is not well known: “It is a totally different kind of nursing from the healthcare one. Of course, it has nothing to do with it. You have to train, spend four months of continuous training, it is demanding, and, at the same time, something very different, very innovative” (E8). The NTCs narrate how not everything can be protocolized, and there is an intangible, uncontrollable part that produces anguish, but at the same time leads to a sense of gratification. The significance of this work is palpable, in order to complete the process of donating an organ: “Around 40% of the transplant coordination is protocolized; 60% are circumstances that change from one situation to another. It is stressful, giving you an adrenaline rush, but then it is very gratifying” (E10). “We are advanced practice nurses. We make finalist decisions. We decide up until when we can continue to offer an organ” (E4).

In nursing care, the management of people and interdisciplinary teams is integrated. This management aspect is inherent to this type of work. Team management means autonomy for the nurse: “Much of it is staff management, surgeons, physicians, staff from other hospitals, it is an enormous world that opens up” (E3). “In this position you have a lot of autonomy, you have to control so many people … it’s different” (E16).

The role of the NTC is the great unknown within hospitals and in the nursing profession itself: “It is a world that few nursing professionals know about. Although there is a lot of talk about it, there is much ignorance of how it works. Above all, our hospital colleagues are unfamiliar with how it works” (E16). “You think that the health personnel are very well trained in everything, but regarding donation, there is no training” (E12).

Furthermore, the work performed by the NTC can change when working outside the hospital, coordinating the out-of-hospital donation process, and when the NTC is integrated into the donation process within the hospital. Thus, during out-of-hospital coordination, the NTC’s work has an important “administrative” workload, through the management of forms and cases, and organizational tasks, although the final objective is always in mind, to successfully obtain the donation: “This work is like business administration. You sit down and start to pick up the phone, it is a very administrative job” (E8). “It fills me with satisfaction that, after all the stress we go through, the sleepless nights and the endless on-call duty, there are people who are getting a transplant and can continue with their lives” (E6). Also, the role of the NTC within the hospital has more contact with the patient and the family, and therefore, the involvement is more direct and closer. The NTC is under great demand, since there is a real possibility that the transplant may not be successful: “When you are with a donor, it is an average of 10–12 h, in which everything goes very fast. It is hard to organize and for everything to run smoothly. It is physically and mentally tough” (E10). “You deal with the drama that the donor’s family faces the death of a loved one. But you have the reward that there are many patients on the waiting list who can benefit from this tragic event” (E14).

### 3.2. Facilitators and Barriers in the NTC´s Job

One of the main facilitators listed by the NTCs was the pride of working within the National Transplant Organization: “You are within an organization that is very well organized. Nationally and internationally recognized. Feeling that you are a member of it gives you a sense of pride” (E3). On all levels, the NTCs recognized that occupying the role of transplant coordinator provides them with a recognized identity within the health system, different from that of a staff/ward nurse. “You are not a number, you are not a ratio, you are not the number two nurse on the oncology floor, there must be three per shift. Here you are, you have your projects, your colleagues know what you do, the management knows what you do and encourages you” (E4). Another facilitator is the specific training offered to start this job. There is a commitment by the organization to training its own staff, the nurse “is part of something else”: “There is exclusive dedication to training with a tutored program. Nurses are not used to training you at the beginning of a new job” (E5).

The NTCs acknowledged that they feel that their work within the transplant organization is known and recognized among the nurses themselves (among their peers), as well as among the remaining health professionals and the general population. Thanks to their efforts, their usefulness and contribution becomes more tangible and noticeable: “You feel that you are treated very well. For society in general and for other professionals. Our function here may be a little more visible” (E9). “The respect that health staff have, the recognition of your good work” (E16).

One of the barriers identified was the great demand for the nurse to become personally involved, which conditions their professional and family life: “On a personal level you become very involved and the family particularly suffers a lot from this type of work because you agree to leave home at 4 in the morning, working on Sundays… For the family, it means that you fail continuously, they are doing worse than us” (E12). “The 24 h shifts are tough; they condition your personal life” (E4). This implication affects the NTCs both emotionally and psychologically: “If you work in a hospital that performs transplants, not only performing the removal of the organs, it is easier to handle because you witness the donation and the transplant to the donor, the success of the entire process. If not, the only thing you see is people in mourning” (E4). “It’s a very stressful job. You have to do a lot of emotional cleansing after a shift, before you get home. You are under a lot of pressure” (E8). The NTCs narrate how this type of work has an expiration date; you cannot work forever, due to the physical and emotional burden: “I think it has a limited duration. It’s tough, even harder than at the special unit level. You don’t have endless time for this emotional and physical burden” (E10).

The NTCs narrated how, being in Spain, a country that is a global reference for the management of donation and transplantation, means that there is a need to update and modify certain criteria, involving continuous efforts, which forces a constant adaptation in how nurses work and provide care: “We have completely changed the profile of the donor and recipient” (E1). “It gets more and more complicated. As transplantation programs have increased, donor requirements have expanded. It is not the same to evaluate a 40-year-old person, as it is normal for them to have no history, than if you are going to evaluate an 80-year-old donor who has a lot of previous pathology” (E12). Despite the fact that the NTCs are proud of their work, in recent times, they have begun to perceive that in the hospitals, quantity predominates, rather than the quality of the care given to donations, and this means that the NTCs do not feel fully recognized within institutions. “The great burden that nursing has here is not valued. Concerning our recognition, we are very much in the shadows, not just economically” (E7).

### 3.3. Not a Job for a Novice Nurse

To work as an NTC requires attitudes and skills developed through experience. This experience is not only focused on nursing, it includes knowing the hospital and how the teams are distributed and function. “You need to possess management knowledge. You have to have a global vision of the hospital. A special sensitivity, a special empathy. Having tools for communication, for team management” (E11). Similarly, it is necessary to have experience in the units from where potential donors, such as ICUs, are “recruited”: “A minimal amount of professional experience after finishing the degree in special units, in generating units: ICU, emergency room, operating room” (E1). Knowledge of the functioning of organ donor generating units helps and facilitates the coordination of the donation and transplantation process, the management of the donor, and the care of the family, and reduces the time required. There is a guideline that is followed, however, nurses must know how to decide when this can be flexible: “We follow the criteria that are described. We act as referees, but sometimes I have to skip those criteria” (E4).

Experience is helpful when handling critical situations. Crisis management means starting the donation process with mourning for the death of a family member, and requesting the disposal of organs and tissues. This is one of the requirements of the coordinator’s job. “The organ donation process is quite specific because it is hard when you have to make a very tough request about the organs and tissues at a time when the family is in post-traumatic shock” (E2). As the NTC, optimal performance during the organ donation process requires motivation, empathy, and communication skills, including the NTC’s ability to self-care and knowing when work can affect you: “You must have the ability to get away from this a little, to recover completely in a very short time, to continue at 200%” (E7).

Finally, the pressure that arises during the donation process must be managed and relieved at all levels. Hospital coordination involves having to care for the donor’s family and surgical teams work under the pressure of respecting the appropriate ischemia times for the transplant: “You have to be able to have a lot of patience and handle a lot of treatment” (E4). “Take care of the forms. We have to be assertive and firm in that sense, calm when coordinating” (E6). “You need to handle things tactfully. We speak with coordinators, with transplant teams, with many people who are experiencing stress and everything falls on you. You have to calm down the situation and shift it to the right place and not face the parts that are currently under pressure” (E8).

### 3.4. NTCs Facing a Paradigm Shift

NTCs are facing a paradigm shift in their role within the complex world of healthcare. Participants spoke of the need to increase their visibility in the scientific community through the generation of evidence in organ donation, along with their role in the generation of qualitative knowledge, in caring for the patient, the family, and the coordination of resources. “More and more, we are assuming greater competencies, fighting for the visibility of nurses. Acquiring more skills, more power, more relevance” (E5).

The NTCs face situations where there is a confrontation of roles and competencies between intensivists, and between the nurses (operating theaters) themselves, which can lead the NTC away from its main objective: taking care of the donor’s family and fulfilling the donation in optimal conditions: “I think that the role of the hospital coordinating nurse is deteriorating. All responsibility for donating and caring for the family is being left to the operating room nurse rather than evaluating patients, potential donors, or talking to the family. The intensivists have kept it. You have to be a doctor to talk with peers…” (E4). “Nursing is going to have to keep fighting. Our case is unusual and you have to continue to demonstrate things that others do not have to demonstrate because they are doctors” (E12). Lastly, in Spain, the donor’s profile has changed. “The shifts are increasingly complex. We have more and more donors. Coordination is complex and to maintain the excellence of the system and what we do, we would have to be more personal” (E6).

## 4. Discussion

The work of the NTC is key for the success of donation and transplant programs, as described in the Spanish model, and other models such as the United Kingdom, where the coordination of the process is carried out by nurses [25]. Within the nursing profession, this is an unknown job among peers, requiring specific training with professional independence within the health system [26]. The transplant coordinator is valued; however, a lack of professional definition and educational preparation has created confusion regarding the role [27]. It is important to include curricular training that addresses the donation of organs in nursing [28]. The authors of the present study believe that NTCs should be considered as Advanced Practice Nursing (APN). Thus, APN is described by two key factors: the expansion of competencies beyond the usual range and an orientation towards the promotion of the profession and its holistic values, focused on health and caring for the person [29]. The Spanish consensus of competencies for APN is based on 12 domains (54 competencies) which include: evidence-based research and practice, clinical leadership and consulting, quality management and clinical safety, and autonomy for professional practice, among others [30,31]. In addition, the training nurses receive conditioned attitudes towards transplants and may influence the number of future transplants [32,33]. For this reason, Allahverdi et al. [32] reported that it is necessary to begin nurses’ university education early on.

NTCs face the coordination of a complex process that involves emotional aspects, resources, team work, and information and security management. Our results have highlighted several facilitating aspects of the process, such as the pride of working for the NTO, which is an institution of national and international recognition. Furthermore, due to the specific nature of the work that is fulfilled, the individual recognition the position occupies in the system is greater than that of a regular nurse. Recently, Becker et al. [34] conducted a study regarding the organ donation process which gathered the expert opinions of intensive care nurses, physicians, transplant coordinators, and transplant surgeons from Austria, Germany, Spain, and the U.K. This report showed how all the participants agreed that doctors need to be supported by full-time in-house special nurses who organize donor evaluation, transport logistics, and pastoral care, if required [34].

Also, previous training was a facilitator together with the social respect within the profession. Another Spanish study has described positive aspects of transplant coordination work. These include the successful outcomes of transplants, the social appreciation of family members (society), and the coordination and quality of interprofessional relationships and recognition of nursing knowledge [14].

In addition, barriers have been described that hinder the development of work. Our results describe emotional barriers mainly in NTCs working in hospitals given the constant interaction with death and grieving family members. Altinisik and Alan [10] described this as compassionate fatigue, traumatic stress resulting in transplant coordinators having stress levels above normal compared to other healthcare workers. Danet Danet et al. [14,15] in their study in southern Spain also concluded that the communication of death is a difficult aspect. Also, compassionate fatigue may be higher in NTCs from nontransplant hospitals compared to transplant hospitals [14]. Our results also concur with those of Danet Danet et al. [14,15] describing the impact on social and family life and the need for “emotional cleansing” to maintain mental health. This study identified institutional barriers regarding the change in the donor profile [14,15]. The need for staff and the perception that quantity prevails over quality in the work of NTCs are aspects which are not fully recognized. Furthermore, it is necessary to point out the existence of patient and family barriers based on cultural and religious influences, which can condition the acceptance of the transplant. In Iran, Abbasi et al. [35] showed how poor knowledge about brain death and organ transplantation from a dead body, cultural beliefs, religious beliefs, deficiencies of the requesting process, fears and concerns, and the inability to make a decision were important barriers to accepting organ donation by families and relatives.

The work fulfilled by NTCs is described as a specialty job, which is not for novices. Communication skills and team management are needed. In addition to knowing the national system, out-of-hospital and intrahospital resources are needed. Hence, the main candidates are nurses who are considered special units, working in emergency departments, ICUs, or operating rooms. In this sense, YazdiMoghaddam et al. [26] reported that ICU nurses must have skills and knowledge that enable them to understand the concept and diagnosis of brain death, without any religious or cultural beliefs opposing organ donation, being well informed of the care process and how to interact with the families. Furthermore, Chen et al. [36] described how nursing managers should have experience in the implementation of evidence-based practice. Also, Zendrato et al. [37] reported the characteristics of an effective manager: communication skills, leadership, education, work duration, and work experience. We can match this description with the needs identified in our study. The need for experienced nurses to fulfill this role is due to the complexity of the process at all levels. Many novice nurse practitioner experienced increased anxiety in the first year of transitioning into practice [38]. Bryant and Parker [39] concluded in their study that nurses who have participated in an educational program assign more value to the work accomplished and have a greater feeling of preparation for practice, and confidence, which increases the level of job satisfaction.

Finally, NTCs are facing a paradigm shift. Concurrently, in the field of donation and transplantation, there are new fields in clinical practice where nursing must be present, such as controlled asystole donation with perfusion in situ and the appearance of preservation programs in normothermia and hypothermia [40]. Moreover, the donor’s profile has changed. In 2019, based on the Spanish Donation and transplantation activity report [2], adult donors had a mean age of 61.5 (SD 14.9), 23.5% of the cadaveric donation in 2019 was between 70 and 79 years old, and 8.5% was over or equal to 80 years old, which involves an increased effort when collecting data from the medical history due to associated comorbidities that can make it difficult to accept organs. In addition, innovation in the field of organ preservation complicates donation logistics by coordinating the transfer of perfusion equipment to small hospitals with fewer resources [41].

Concerning limitations, it is important to consider a possible influence of the Spanish cultural context on the themes generated by this research [42]. Since the approval of the donation and transplantation law in 1979, the Spanish population has generously and selflessly cooperated in the organ donation process. This law placed the Spanish population in an opt-out or presumed consent situation [1]. In Spain, Matesanz et al. [43] reported how the success rates in Spanish derive from a specific organizational approach to ensure the systematic identification of opportunities for organ donation and their transition to actual donation, and the promotion of public support for organ donation after death. In the health system organization, one of the most influential factors for the general population is the attitude of health professionals according to previous studies, which report that the attitude of nurses towards organ donation is widely favorable [44].

## 5. Conclusions

Finally, NTCs described their job as a different type of job for experienced nurses, increasing skills and responsibilities and characterized as autonomous and based on scientific research. Facilitators include high education and training, pride of being part of the Spanish NTO, and the social and professional recognition of helping to accomplish the transplant. NTCs encounter emotional/psychological and institutional barriers. NTCs are facing a paradigm shift due to changes in society and innovation in organ preservation. Finally, NTCs oversee and cater to the quality of the donation and transplant process, beyond mere statistical reports.

Regarding implications for practice, our results highlight the need to enhance the positive aspects within transplant coordination, being able to implement training courses to educate more nurses who can share and thus reduce the workload. In addition, the role of the NTC as an advanced practice nurse should be recognized, thus promoting professional recognition and professional progression within transplant coordination.

## Figures and Tables

**Table 1 ijerph-17-07996-t001:** Semistructured interview guide.

Investigated Theme	Questions
National transplant organization experience	Tell me about your experience and professional skills as a donation and transplant coordinator.
Barriers	Could you identify relevant barriers of being a coordinator?
Job description	What does a nurse need to be a donation and transplant coordinator?
Future perspectives	How do you see the future of nursing within the process of donation and transplants?

**Table 2 ijerph-17-07996-t002:** Example of the inductive coding process analysis.

Narratives	Codes	Code Groups	Theme
“From a professional point of view, it is quite rewarding. You have a professional projection that you do not have in other areas” (E9)	Development and innovation	Nursing professional development factors within transplant coordination	Facilitators and barriers for an NTC´s job.
“You move around knowing a lot of people, all in the field of donation and transplantation, but you meet great professionals who also have a lot of responsibility and who are at the forefront of their business and that seems like a privilege to me” (E6)	Pride and prestige		
“There is nothing to validate our knowledge. When it is something so specific and everchanging, we are not at all recognized” (E10)	Lack of professional recognition	Negative experiences and aspects of the NTC’s job	
“The 24 h shifts are becoming more complex. We have more and more donors. Coordination is more complex” (E6)	Labor complexity		
“The usual barriers. We have a low and poor concept of ourselves. We have low self-esteem” (E5)	Negative self-concept		

**Table 3 ijerph-17-07996-t003:** Trustworthiness criteria applied.

Criteria	Techniques Performed and Application Procedures
Credibility	Investigator triangulation: each data source was analyzed. Thereafter, team meetings were performed during which the analyses were compared and themes were identified. Triangulation of data collection methods: including semistructured interviews and researcher field notes. Participant validation: this consisted of asking the participants to confirm the data obtained at the stages of data collection.
Transferability	In-depth descriptions of the study performed, providing details of the characteristics of researchers, participants, contexts, sampling strategies, and the data collection and analysis procedures.
Dependability	Audit by an external researcher: an external researcher assessed the study research protocol, focusing on aspects concerning the methods applied and the study design.
Confirmability	Investigator triangulation, data collection triangulation. Researcher reflexivity was encouraged via the previous positioning, performance of reflexive reports, and by describing the rationale behind the study.

**Table 4 ijerph-17-07996-t004:** Findings.

Findings	Description
A different job for nurses	The NTC’s work is described as an autonomous and demanding role with the need for specialized training and the ability to coordinate multidisciplinary teams.
Facilitators and barriers of the NTC´s job	The specialized training and the pride of being part of the NTO facilitate the work of the NTC. Moreover, the duration and intensity of the donation process, as well as the institutional pressure, are described as barriers.
Not a job for a novice nurse	The role of the NTC requires work experience in managing multidisciplinary teams and adequate communication in stressful situations, as well as handling crisis situations with family members.
NTCs facing a paradigm shift	The profile of the donor has changed in Spain, causing the NTC to face more complex coordination, developing skills and demanding professional visibility and recognition.

NTC: nurse transplant coordinator.

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
