# Peer review of "Facilitators and Barriers in the Organ Donation Process: A Qualitative Study among Nurse Transplant Coordinators"

_ijerph, 2020, doi:10.3390/ijerph17217996_

Round 1

Reviewer 1 Report

This is a phenomenological study on the lived experiences of nurse transplant coordinators, the topic is interesting and important, and the study appears properly implemented. All essential details about the study are included, and the findings are reasonable. No major issue identified except the English language and style will benefit from help from a professional copy-editor.

Author Response

RESPONSE LETTER

ijerph-945582

Old Title: “It seems easy, but it is not”. Facilitators and barriers in the organ donation process: a qualitative study.

Modified title after review process: Facilitators and Barriers in the Organ Donation Process: A Qualitative Study among Nurse Transplant Coordinators

 We would like to thank the Editors and the Reviewers for their careful consideration of our manuscript. We would also like to thank the Reviewers’ suggestions, which we believe have enhanced the quality of the manuscript. We have highlighted all the changes we have made throughout the text. Below please find a detailed list of how we have addressed each comment.

REVIEWER 1 

Suggestions for Authors

This is a phenomenological study on the lived experiences of nurse transplant coordinators, the topic is interesting and important, and the study appears properly implemented. All essential details about the study are included, and the findings are reasonable. No major issue identified except the English language and style will benefit from help from a professional copy-editor.

Response: We have followed the reviewer´s recommendation. The manuscript has been reviewed by a native English speaker from a professional editing company. As a result extensive changes have been made throughout the text to improve the quality of the manuscript. Please see proofreading certificate attached.  

Reviewer 2 Report

This is an interesting study that focuses on the qualitative evaluation of the organ donation process.

Title: relevant, in my opinion, the first sentence is redundant.

Introduction: clear, specific and informative.  Good epidemiological description focused on the country and region of the study. Although the authors describe the role of the transplant nurse coordinator (TNC), their specific functions within the multidisciplinary team should be clarified for the readers to have a better understanding. I encourage the authors to justify more clearly the need for their study, and how it differs from others like (Danet Danet et al, 2019).

Objectives: well thought out.

Methodology: Qualitative/phenomenological design, a theoretical-philosophical framework is missing as the basis of the study. Table 1. (The positioning of the researchers) should be cited in the text.  The context of the study (p. 3, l. 88-99) should focus on this research investigation. The authors give a general description that should be in the introduction/background.  Why weren’t focus groups used in the data collection? Table 2 (semi-structured interview guide) should be more explicit.

Rigor: How were discrepancies solved in the triangulation process of the researchers? Were topics excluded due to a lack of consensus?

Results: very interesting, clear and direct.  The experience of the participants is noted, indicating new points of view on roles, functions, facilitators and barriers in the process. I commend the authors for the clarity of the testimonials presented. Add standard deviation to means of age, years of experience and work in transplant coordination (P.5, L. 156-159). Add an example of the inductive coding process (codes-subtopics-topics). Incorporate a summary table of the results to increase visibility and make it easier to read.

Discussion: well developed. Some areas lack greater contrast with results in other countries/health systems. 

Limitations: what are the authors referring to with the influence of the Spanish culture in the topics generated by the research?

Conclusions: clear and concrete.  Implications for practice are missing. It is a good piece of work, I hope the proposed modifications can improve it.

Author Response

RESPONSE LETTER

ijerph-945582

Old Title: “It seems easy, but it is not”. Facilitators and barriers in the organ donation process: a qualitative study.

Modified title after review process: Facilitators and Barriers in the Organ Donation Process: A Qualitative Study among Nurse Transplant Coordinators

We would like to thank the Editors and the Reviewers for their careful consideration of our manuscript. We would also like to thank the Reviewers’ suggestions, which we believe have enhanced the quality of the manuscript. We have highlighted all the changes we have made throughout the text. Below please find a detailed list of how we have addressed each comment.

REVIEWER 2

This is an interesting study that focuses on the qualitative evaluation of the organ donation process.

Title: relevant, in my opinion, the first sentence is redundant.

Response: We have edited the title removing the redundant sentence.

Introduction: clear, specific and informative.  Good epidemiological description focused on the country and region of the study. Although the authors describe the role of the transplant nurse coordinator (TNC), their specific functions within the multidisciplinary team should be clarified for the readers to have a better understanding.

Response: We have edited the text following the reviewer´s suggestion.

We have included the following in the introduction:

Spanish Nurse Transplant Coordinators (NTCs) play a significant role in the NTO, particularly in relation to the coordination of organ donation and transplantation, data and information management, dissemination, training and research [5]. According to the law, the duties of the NTO include procurement coordination, national distribution and international exchange of organs, tissues and cells for transplantation [6]. The main duty of the hospital transplant coordination care unit is the organization and optimization of the collection and clinical use of human organs [6]. Likewise, the role of the NTC in the Regional Office for Transplant Coordination (ROTC) [1] is to: “collaborate in the fulfillment of general objectives set by the Commission of Transplants of the Inter-territorial Council, and coordinate actions at the hospital level and at the level of the targeted population for the promotion of donation and transplantation of organs and tissues… they also assume the coordination of resources, information, dissemination, promotion, continuing education, as well as research and cooperation with other institutions”(p.42)

I encourage the authors to justify more clearly the need for their study, and how it differs from others like (Danet Danet et al, 2019).

Response: We have edited the text accordingly.

The following text has been included:

Our study gathers the exclusive experience of nursing professionals working as transplant coordinators. In addition, this study seeks to understand the global experience of the donation process, which is why participants were included from hospital transplant coordination units, as well as regional and national coordinators in Madrid that perform out-hospital coordination activity. Unlike previous studies such as the paper by Danet Danet et al [14,15], the purpose of the present study was to describe the experience of nurses transplant coordinators in the process of organ donation. Additionally, this study sought to analyze the current barriers and facilitators faced by NTCs in the process of organ donation.

We have included the following reference: 15. Danet Danet, A.; Jimenez Cardoso, P.M. Emotional experiences of health professionals in organ procurement and transplantation. A systematic review. Cir Esp. 2019;97(7):364-376. doi:10.1016/j.ciresp.2019.01.015

Objectives: well thought out.

Response: Thank you for this comment.

Methodology: Qualitative/phenomenological design, a theoretical-philosophical framework is missing as the basis of the study.

Response: We agree with the reviewer, we have included further information on the design and the type of phenomenology school used (Husserl, descriptive).

We have included this information in the methods section:

A qualitative phenomenological descriptive design was conducted following Husserl´s framework [17]. In the field of qualitative studies, phenomenology attempts to understand how individuals construct their world view. Thus, it looks through a window into other people’s experiences. Phenomenology attempts to identify the essence of participants’ lived experiences, which is the subjective reflection on human beings when taking part in events in a specific space and time [18]. The Husserl’s framework [19,20] guided this study. For Husserl, the aim of phenomenology is the study of aspects as they appear to reach an essential understanding of human experience [19,20]. To consider subjective experiences, the researcher assumes a certain attitude of attentive openness and readiness for a proper understanding of the unique meaning of participants’ lived-experiences [21]. This experience always has a meaning for the person who has undergone the same, and thus, phenomenological research uses first-person narratives from the participants themselves as a data source [17,21].

We have included new references:

  1. Phillips-Pula, L.; Strunk, J.; Pickler, R.H. Understanding phenomenological approaches to data analysis. J Pediatr Health Care 2011;25: 67-71. doi: 10.1016/j.pedhc.2010.09.004.
  2. Dowling, M. From Husserl to van Manen. A review of different phenomenological approaches. Int J Nurs Stud. 2007;44(1):131-42. doi: 10.1016/j.ijnurstu.2005.11.026.
  3. Norlyk, A.; Harder, I. What makes a phenomenological study phenomenological? An analysis of peer-reviewed empirical nursing studies. Qual Health Res. 2010;20(3):420-31. doi: 10.1177/1049732309357435.
  4. Carpenter, C.; Suto, M. (2008). Qualitative research for occupational and physical therapist: A practical guide. Blackwell: Oxford, England, 2008.

Table 1. (The positioning of the researchers) should be cited in the text. 

Response: We have followed the reviewer’s suggestion. We have removed table 1 and included this content in the manuscript text.

New text:

Researchers based their approach on an interpretivist paradigm. This paradigm was based on the assumption that human beings construct their own social reality, and that knowledge is built through increasingly nuanced reconstructions of individual experiences. Also, researchers believed that the nurse transplant coordinator plays a significant role in the Spanish model for donation and transplantation. The care of the deceased donor is the key to the success of the donation process. Finally, researchers believed that nursing care during the organ donation process would be able to define improvements and advances that positively influence the increase in the donor pool.

The context of the study (p. 3, l. 88-99) should focus on this research investigation. The authors give a general description that should be in the introduction/background. 

Response: We have followed the reviewer’s suggestion. We have removed content and included it in the introduction section. Also, we have included new information regarding the current context focused on research investigation.

The Spanish public health system centralizes and directs organ donation. This requires coordination between viable donors, the family, and the donation process. This donation process can be subject to stressors and barriers that delay donation. In this context, the role of the NTC is fundamental to eliminate barriers, and empower facilitators that allow donation. However, the NTC can be subject to professional difficulties in public hospitals, such as the lack of recognition, or the lack of its own space within the donation process.

Why weren’t focus groups used in the data collection?

Response: Based on the research question, the researchers chose phenomenology as the design. This qualitative design recommends in-depth interviews as fundamental data collection tools, individually for each participant. This is the reason for not holding focus groups.

In this manner, Moser & Korstjens (2018) reported that interviews are suitable to phenomenology but not focus groups as a data collection tool. See figure (box 2) obtained from Moser A, Korstjens I. Series: Practical guidance to qualitative research. Part 3: Sampling, data collection and analysis. Eur J Gen Pract. 2018 Dec;24(1):9-18. doi: 10.1080/13814788.2017.1375091. Epub 2017 Dec 4. PMID: 29199486; PMCID: PMC5774281.

Also, Carpenter & Suto (2008) described: “In phenomenological research, the aim is to gain rich and thick information (…) Data are collected through one or two extensive semi-structured interviews with each participant.” (p.66).

References: Carpenter, C., & Suto, M. (2008). Qualitative research for occupational and physical therapist: A practical guide. Oxford, England: Blackwell.

Table 2 (semi-structured interview guide) should be more explicit.

Response: The authors disagree on this point. In phenomenology, it is necessary to ask open questions so that the participants are able to narrate their experience from their perspective. Therefore researchers should use a question guide that suggests or directs the interview but does not condition the content. From our perspective, more extensive questions could have led the interview away from what was relevant to the participant's experience.

However, the authors believe that it is necessary to explain how the interview took place, in order to understand how, through open questions, other content could be identified and deepened.

We have now included:

Subsequently, the researchers listened carefully, noting the key words and topics identified in the participants’ responses and using their answers to ask further questions for clarification. In this manner, relevant information was collected from the participants’ perspective [21]. Additionally, the researchers used prompts: (a) to encourage the participants to go into further detail (‘Can you expand more on that?’), (b) to encourage the participant to continue talking (‘Have you experienced the same thing since?’), and (c) to dissipate confusion (paraphrasing something that the patient had said) [21].

We have included a new reference: 21.   Carpenter, C.; Suto, M. (2008). Qualitative research for occupational and physical therapist: A practical guide. Blackwell: Oxford, England, 2008.

Rigor: How were discrepancies solved in the triangulation process of the researchers? Were topics excluded due to a lack of consensus?

Response: In this study, no content was removed due to lack of consensus. The results were redefined, and there were changes in the groupings of codes, and the way of naming or stating the final themes.

Results: very interesting, clear and direct.  The experience of the participants is noted, indicating new points of view on roles, functions, facilitators and barriers in the process. I commend the authors for the clarity of the testimonials presented.

Response: Thank you for this comment.

Add standard deviation to means of age, years of experience and work in transplant coordination (P.5, L. 156-159).

Response: Done.

Add an example of the inductive coding process (codes-subtopics-topics).

Response: Edited as suggested.

We have included a new table:

Table 2. Example of the inductive coding process analysis

Narratives

Codes

Code groups

Theme

From a professional point of view, it is quite rewarding. You have a professional projection that you do not have in other areas” (E9)

Development and innovation.

Nursing professional development factors within transplant coordination.

Facilitators and barriers for a NTC´s job.

You move around knowing a lot of people, all in the field of donation and transplantation, but you meet great professionals who also have a lot of responsibility and who are at the forefront of their business and that seems like a privilege to me” (E6)

Pride and prestige.

There is nothing to validate our knowledge. When it is something so specific and everchanging, we are not at all recognized” (E10)

Lack of professional recognition.

Negative experiences and aspects of the NTC’s job.

The 24 hours shifts are becoming more complex. We have more and more donors. Coordination is more complex” (E6)

Labor Complexity

The usual barriers. We have a low and poor concept of ourselves. We have low self-esteem" (E5)

Negative self-concept

Incorporate a summary table of the results to increase visibility and make it easier to read.

Response: We have made changes as suggested.

We have included a new table:

 Table 4. Findings

Findings

Description

A different job for nurses

The NTC's work is described as an autonomous and demanding role with the need for specialized training and the ability to coordinate multidisciplinary teams.

Facilitators and barriers of the NTC´s job

The specialized training and the pride of being part of the NTO facilitate the work of the NTC. Moreover, the duration and intensity of the donation process, as well as the institutional pressure are described as barriers.

Not a job for a novice nurse

The role of the NTC requires work experience in managing multidisciplinary teams, adequate communication in stressful situations, as well as handling crisis situations with family members.

NTCs facing a paradigm shift

The profile of the donor has changed in Spain, causing the NTC to face more complex coordination, developing skills and demanding professional visibility and recognition.

Discussion: well developed. Some areas lack greater contrast with results in other countries/health systems. 

Response: Thank you for this comment. We have rewritten the discussion section and we now include new references to compare our findings with other countries. On the other hand, it is difficult to find specific jobs for nurse transplant coordinators in other countries, because this figure is typical of the Spanish organ donation system. Hence, the originality and relevance of the study.

We have included new references:

  1. YazdiMoghaddam H, Manzari ZS, Mohammadi E. Nurses' Challenges in Caring for an Organ Donor Brain Dead Patient and their solution strategies: A Systematic Review. Iran J Nurs Midwifery Res. 2020 Jun 17;25(4):265-272. doi: 10.4103/ijnmr.IJNMR_226_18.
  2. Allahverdi TD, Allahverdi E, AkkuÅŸ Y. The Knowledge of Nursing Students About Organ Donation and the Effect of the Relevant Training on Their Knowledge. Transplant Proc. 2020 Jun 28:S0041-1345(19)31472-1. doi: 10.1016/j.transproceed.2020.04.1815.
  3. Haddiya I, El Meghraoui H, Bentata Y, Guedira M. Attitudes, Knowledge, and Social Perceptions toward Organ Donation and Transplantation in Eastern Morocco. Saudi J Kidney Dis Transpl. 2020 Jul-Aug;31(4):821-825. doi: 10.4103/1319-2442.292316.
  4. Abbasi P, Yoosefi Lebni J, Nouri P, Ziapour A, Jalali A. The obstacles to organ donation following brain death in Iran: a qualitative study. BMC Med Ethics. 2020 Sep 1;21(1):83. doi: 10.1186/s12910-020-00529-8.
  5. Matesanz R, Domínguez-Gil B, Coll E, Mahíllo B, Marazuela R. How Spain Reached 40 Deceased Organ Donors per Million Population. Am J Transplant. 2017 Jun;17(6):1447-1454. doi: 10.1111/ajt.14104

Limitations: what are the authors referring to with the influence of the Spanish culture in the topics generated by the research?

Response: We have followed the reviewer´s suggestion. We have added more information in the limitations section regarding the influence of the Spanish culture.

Since the approval of the donation and transplantation law in 1979, the Spanish population has generously and selflessly cooperated in the organ donation process. This law placed the Spanish population in an opt-out or presumed consent situation [1]. One of the most influential factors in the general population is the attitude of health professionals and previous studies, which report that the attitude of nurses towards organ donation is widely favorable [38].

We have included a new reference: 38.  Montero Salinas A.; Martínez-Isasi S.; Fieira Costa E.; Fernández García A.; Castro Dios D. J.; Fernández García D. Knowledge and attitudes toward organ donation among health professionals in a third level hospital. Rev Esp Salud Publica. 2018;92:e201804007.

Conclusions: clear and concrete.  Implications for practice are missing. It is a good piece of work, I hope the proposed modifications can improve it.

Response: We have followed the reviewer´s suggestion, adding more information in the conclusions section regarding Implications for practice:

Regarding implications for practice, our results highlight the need to enhance the positive aspects within transplant coordination, being able to implement training courses to educate more nurses who can share and thus reduce the workload. In addition, the role of the NTC as an advanced practice nurse should be recognized, thus promoting professional recognition and professional progression within transplant coordination.

We hope that you are satisfied with this revision and that the manuscript is now suitable for publication in IJERPH.

Sincerely,

The Authors

Reviewer 3 Report

This paper presents an important study on the experience of organ transplant coordinators in Spain.

In general the document is well structured,  but some modifications are necessary to guarantee methodological coherence.

1. Title: the title of the paper should be revised as it does not make explicit reference to the figure under study. The title proposed by the authors may cause confusion.It would be advisable to incorporate the word coordinator in the title because it is the object of study.

2. Design of the research. The authors say that they employ a phenomenological perspective but which type of phenomenological approach did you use? The type used is a critical factor to how you approach your analysis. Authors should describe their understanding of the phenomenological methodology and how they used it.
There is tension in phenomenological studies between appropriate categorization and categories that emerge and are uncovered from the data.

3. Results. Depending on the type of phenomenological approach used, I would expect more depth to interpretations of what the data/quotes meant.

4. Discussion. I do not fully understand the origin of this assertion" Our results reported consideration of NTC as Advanced Practice Nursing (APN)". I believe that it is an interpretation of the researchers without a basis in the results and that it would be interesting to study it more deeply in future research.

5.With regard to the limitations of the study, I do not believe that the generalization of the results to other populations is a limitation, because the main characteristic of qualitative research is the study of the local5. With regard to the limitations of the study, I do not believe that the generalization of the results to other populations is a limitation, because the main characteristic of qualitative research is the development of the local and specific.

Author Response

RESPONSE LETTER

ijerph-945582

Old Title: “It seems easy, but it is not”. Facilitators and barriers in the organ donation process: a qualitative study.

Modified title after review process: Facilitators and Barriers in the Organ Donation Process: A Qualitative Study among Nurse Transplant Coordinators

We would like to thank the Editors and the Reviewers for their careful consideration of our manuscript. We would also like to thank the Reviewers’ suggestions, which we believe have enhanced the quality of the manuscript. We have highlighted all the changes we have made throughout the text. Below please find a detailed list of how we have addressed each comment.

REVIEWER 3

This paper presents an important study on the experience of organ transplant coordinators in Spain. In general the document is well structured,  but some modifications are necessary to guarantee methodological coherence.

  1. Title: the title of the paper should be revised as it does not make explicit reference to the figure under study. The title proposed by the authors may cause confusion. It would be advisable to incorporate the word coordinator in the title because it is the object of study.

Response: We have followed the reviewer´s comment.

The revised title is as follows: “Facilitators and Barriers in the Organ Donation Process: A Qualitative Study among Nurse Transplant Coordinators.”

  1. Design of the research. The authors say that they employ a phenomenological perspective but which type of phenomenological approach did you use? The type used is a critical factor to how you approach your analysis. Authors should describe their understanding of the phenomenological methodology and how they used it.

Response: We agree with the reviewer, we now include further information on the design and the type of phenomenology school used (Husserl, descriptive) and we justify the suitability of using the analysis proposal used (systematic text condensation) with the design used (phenomenology).

We have included this information in the method section:

A qualitative phenomenological descriptive design was conducted following Husserl´s framework [17]. In the field of qualitative studies, phenomenology attempts to understand how individuals construct their world view. Thus, it looks through a window into other people’s experiences. Phenomenology attempts to identify the essence of participants’ lived experiences, which is the subjective reflection on human beings when taking part in events in a specific space and time [18]. The Husserl’s framework [19,20] guided this study. For Husserl, the aim of phenomenology is the study of aspects as they appear to reach an essential understanding of human experience [19,20]. To consider subjective experiences, the researcher assumes a certain attitude of attentive openness and readiness for a proper understanding of the unique meaning of participants’ lived-experiences [21]. This experience always has a meaning for the person who has undergone the same, and thus, phenomenological research uses first-person narratives from the participants themselves as a data source [17,21].

Below, we have developed the justification regarding the appropriateness of the analysis proposal used (systematic text condensation) with the design used (descriptive phenomenology, Husserl).

On the other hand, the phenomenology by Husserl uses the phenomenological reduction for analyzing and describing the experience of people who live a certain phenomenon. Amadeus Giorgi, psychologist who uses the proposal by Husserl (Phillips-Pula et al., 2011), together with other authors, describe a series of steps for phenomenological analysis (Giorgi, 2000, 2005). These steps consist in describing the experience, identifying each unit, in order to transform these into meanings that express and describe the experience (Giorgi, 2005; Phillips-Pula et al., 2011). In this analysis proposal, the description and identification of the units that constitute the same is fundamental (Dowling, 2007; Phillips-Pula et al., 2011).

“All three psychologists (van Kaam, Giorgi and Colazzi) employ a similar series of steps: (a) the original descriptions are divided into units, (b) the units are transformed by the researcher into meanings that are expressed in psychological and phenomenological concepts and (c) these transformations are combined to create a general description of the experience.” (Dowling, 2007.p. 135).

“Giorgi exemplifies his method by a study of a learning experience, containing a four-step Procedure (…) 1. One reads the entire description in order to get a general sense of the whole statement. 2. Once the sense of the whole has been grasped, the researcher goes back to the beginning and reads through the text once more with the specific aim of discriminating “meaning units” from within a psychological perspective, with a focus on the phenomenon being researched. 3. Once the meaning units have been delineated, the researcher goes through all of the meaning units and expresses the psychological insight contained in them more directly. This is especially true of the meaning units most revelatory of the phenomenon under consideration. 4. Finally, the researcher synthesizes all of the transformed meaning units into a consistent statement regarding the subject’s experience.” (Malterud, 2012.p.796)

On the other hand, Giorgi continues to use the philosophical framework built by Husserl to continue developing the application of phenomenology to the field of care (Giorgi, 2000; 2005). Therefore, there is a relationship between the studies by Husserl on phenomenology and the Works by Giorgi and its application to research. Besides, Giorgi’s proposal has been used as a method of analysis for phenomenological studies in nursing.

“Of all the phenomenological psychologists, Amedeo Giorgi continues to write regularly about phenomenology as a method for the human sciences (…) His human science approach to phenomenology follows a rigorous program of Husserl’s writings and maintains that the object of phenomenological description is achieved ‘‘solely’’ through a direct grasping of the essential structure of phenomena (...) Giorgi’s influence on nursing theory is also evident in Watson’s theory of caring, where she elaborates on the notion of a ‘‘human science’’ proposed by the followers of the Duquesne school of phenomenology (…) Giorgi’s work is utilised frequently by nurse researchers in the analysis of interview data.”.(Dowling, 2007.p. 135).

Based on Giorgi’s proposal, within descriptive phenomenology (Husserl), Maltereud (2012) proposed a new proposal for the analysis of lived experience: Systematic text condensation.

“Giorgi's psychological phenomenological analysis is the point of departure and inspiration for systematic text condensation (…) Systematic text condensation is a descriptive and explorative method (…) The procedure consists of the following steps: 1) total impression - from chaos to themes; 2) identifying and sorting meaning units - from themes to codes; 3) condensation - from code to meaning; 4) synthesizing - from condensation to descriptions and concepts.” (Malterud, 2012.p.795)

This analysis proposal is based on Giorgi’sthe principles of analysis, which is still a descriptive analysis of the lived experience of the participants in the first person, and which preserves examples from peoples’ life worlds, using accounts from participants to justify the analysis. In this system, decontextualization and recontextualization of data is still used, and the process of bracketing is applied to retain the researcher’s beliefs and their influence on the phenomenon under study.

Systematic text condensation: “ (…) is an elaboration of Giorgi’s principles, including four comparable steps of analysis (…) is also a descriptive approach, presenting the experience of the participants as expressed by themselves, rather than exploring possible underlying meaning of what was said. Following Giorgi, Systematic text condensation holds an explorative ambition to present vital examples from peoples’ life worlds, not to cover the full range of potential available phenomena. A limited number of participants or accounts provides sufficient data for analysis (…) Like Giorgi’s method, Systematic text condensation implies analytic reduction with specified shifts between decontextualization and recontextualization of data.” (Malterud, 2012.p.796).

For this reason, the authors believe that using Systematic text condensation, as a proposal for the analysis of data in a descriptive phenomenological study is coherent as it is based on the proposal of descriptive analysis (Amadeus Giorgi), which is applied for descriptive phenomenological studies, based on Husserl.

References:

  • Dowling, M. (2007). From Husserl to van Manen. A review of different phenomenological approaches. International Journal of Nursing Studies, 44, 131-42.
  • Giorgi, A. (2005). The phenomenological movement and research in the human sciences. Nursing Science Quarterly, 18, 75- 82.
  • Giorgi, A. P. (2000). Concerning the application of phenomenology to caring research. Scandinavian Journal of Caring Science, 14, 11–15.
  • Malterud, K. (2012). Systematic text condensation: a strategy for qualitative analysis. Scandinavian Journal of Public Health, 40, 795-805.
  • Phillips-Pula, L., Strunk, J., & Pickler, R.H. (2011). Understanding phenomenological approaches to data analysis. Journal of Pediatric Health Care, 25, 67-71. doi: 10.1016/j.pedhc.2010.09.004.

However, we acknowledge the need to provide improved clarity to the analysis section in order to offer a better description of the analysis process.

We have included the following text in the analysis section:

Systematic text condensation (STC) is an elaboration of Giorgi’s principles (a follower of Husserl) [18], including four comparable steps of analysis [22]. In this descriptive approach, presenting the experience of the participants as expressed by themselves, and following Giorgi, STC provides an explorative proposal to present vital examples from peoples’ life worlds [22]. Also, like Giorgi’s method [19,20,23,24], STC implies analytic reduction with decontextualization and recontextualization of data [22]. This procedure consists of the following steps [22]: 1) reading all the material and an overview of the data; 2) reviewing the transcript line by line to identify meaning units. A meaning unit is a text fragment containing some information about the research question. Subsequently, coding begins (decontextualization), which includes identifying, classifying, and sorting meaning units and marking these with a code –a label that connects related meaning units into a code group; 3) implying the systematic abstraction of meaning units within each of the code groups established in the second step of analysis. Empirical data are reduced to a decontextualized selection of meaning units sorted as thematic code groups across individual participants; 4) data are reconceptualized, putting the pieces together again. Themes are developed, providing stories that reflect the participants’ experiences [22]).

Also, we have included new references:

  1. Phillips-Pula, L.; Strunk, J.; Pickler, R.H. Understanding phenomenological approaches to data analysis. J Pediatr Health Care 2011;25: 67-71. doi: 10.1016/j.pedhc.2010.09.004.
  2. Dowling, M. From Husserl to van Manen. A review of different phenomenological approaches. Int J Nurs Stud. 2007;44(1):131-42. doi: 10.1016/j.ijnurstu.2005.11.026.
  3. Norlyk, A.; Harder, I. What makes a phenomenological study phenomenological? An analysis of peer-reviewed empirical nursing studies. Qual Health Res. 2010;20(3):420-31. doi: 10.1177/1049732309357435.
  4. Malterud, K. Systematic text condensation: a strategy for qualitative analysis. Scand J Public Health 2012, 40,795-805. doi: 10.1177 / 1403494812465030.
  5. Giorgi, A. (2005). The phenomenological movement and research in the human sciences. Nurs Sci Q. 2005;18(1):75-82. doi: 10.1177/0894318404272112.
  6. Giorgi, A. P. (2000). Concerning the application of phenomenology to caring research. Scand J Caring Sci. 2000;14(1):11-5. doi: 10.1111/j.1471-6712.2000.tb00555.x.

There is tension in phenomenological studies between appropriate categorization and categories that emerge and are uncovered from the data.

Response: From our point of view, we disagree with the reviewer. As we have pointed out in the previous question, there is a coherence between the chosen design (descriptive phenomenology-Husserl), the theoretical framework (interpretivist paradigm) and the analysis proposal used (Systematic text condensation).

We have developed the justification regarding the appropriateness of the analysis proposal used (systematic text condensation) with the design used (descriptive phenomenology, Husserl). For this reason, we believe that our results are congruent with analysis applied.

The phenomenology by Husserl uses the phenomenological reduction for analyzing and describing the experience of people who live a certain phenomenon. Amadeus Giorgi, psychologist who uses the proposal by Husserl (Phillips-Pula et al., 2011), together with other authors, describe a series of steps for phenomenological analysis (Giorgi, 1997, 2000, 2005). These steps consist in describing the experience, identifying each unit, in order to transform these into meanings that express and describe the experience (Giorgi, 1997, 2005; Phillips-Pula et al., 2011). In this analysis proposal, the description and identification of the units that constitute the same is fundamental (Dowling, 2007; Phillips-Pula et al., 2011).

“All three psychologists (van Kaam, Giorgi and Colazzi) employ a similar series of steps: (a) the original descriptions are divided into units, (b) the units are transformed by the researcher into meanings that are expressed in psychological and phenomenological concepts and (c) these transformations are combined to create a general description of the experience.” (Dowling, 2007.p. 135).

“Giorgi exemplifies his method by a study of a learning experience, containing a four-step Procedure (…) 1. One reads the entire description in order to get a general sense of the whole statement. 2. Once the sense of the whole has been grasped, the researcher goes back to the beginning and reads through the text once more with the specific aim of discriminating “meaning units” from within a psychological perspective, with a focus on the phenomenon being researched. 3. Once the meaning units have been delineated, the researcher goes through all of the meaning units and expresses the psychological insight contained in them more directly. This is especially true of the meaning units most revelatory of the phenomenon under consideration. 4. Finally, the researcher synthesizes all of the transformed meaning units into a consistent statement regarding the subject’s experience.” (Malterud, 2012.p.796)

On the other hand, Giorgi continues to use the philosophical framework built by Husserl to continue developing the application of phenomenology to the field of care (Giorgi, 2000; 2005). Therefore, there is a relationship between the studies by Husserl on phenomenology and the Works by Giorgi and its application to research. Besides, Giorgi’s proposal has been used as a method of analysis for phenomenological studies in nursing.

“Of all the phenomenological psychologists, Amedeo Giorgi continues to write regularly about phenomenology as a method for the human sciences (…) His human science approach to phenomenology follows a rigorous program of Husserl’s writings and maintains that the object of phenomenological description is achieved ‘‘solely’’ through a direct grasping of the essential structure of phenomena (...) Giorgi’s influence on nursing theory is also evident in Watson’s theory of caring, where she elaborates on the notion of a ‘‘human science’’ proposed by the followers of the Duquesne school of phenomenology (…) Giorgi’s work is utilised frequently by nurse researchers in the analysis of interview data.”.(Dowling, 2007.p. 135).

Based on Giorgi’s proposal, within descriptive phenomenology (Husserl), Maltereud (2012) proposed a new proposal for the analysis of lived experience: Systematic text condensation.

“Giorgi's psychological phenomenological analysis is the point of departure and inspiration for systematic text condensation (…) Systematic text condensation is a descriptive and explorative method (…) The procedure consists of the following steps: 1) total impression - from chaos to themes; 2) identifying and sorting meaning units - from themes to codes; 3) condensation - from code to meaning; 4) synthesizing - from condensation to descriptions and concepts.” (Malterud, 2012.p.795)

This analysis proposal is based on Giorgi’sthe principles of analysis, which is still a descriptive analysis of the lived experience of the participants in the first person, and which preserves examples from peoples’ life worlds, using accounts from participants to justify the analysis. In this system, decontextualization and recontextualization of data is still used, and the process of bracketing is applied to retain the researcher’s beliefs and their influence on the phenomenon under study.

Systematic text condensation: “ (…) is an elaboration of Giorgi’s principles, including four comparable steps of analysis (…) is also a descriptive approach, presenting the experience of the participants as expressed by themselves, rather than exploring possible underlying meaning of what was said. Following Giorgi, Systematic text condensation holds an explorative ambition to present vital examples from peoples’ life worlds, not to cover the full range of potential available phenomena. A limited number of participants or accounts provides sufficient data for analysis (…) Like Giorgi’s method, Systematic text condensation implies analytic reduction with specified shifts between decontextualization and recontextualization of data.” (Malterud, 2012.p.796).

For this reason, the authors believe that using Systematic text condensation, as a proposal for the analysis of data in a descriptive phenomenological study is coherent as it is based on the proposal of descriptive analysis (Amadeus Giorgi), which is applied for descriptive phenomenological studies, based on Husserl.

References:

  • Dowling, M. (2007). From Husserl to van Manen. A review of different phenomenological approaches. International Journal of Nursing Studies, 44, 131-42.
  • Giorgi, A. (2005). The phenomenological movement and research in the human sciences. Nursing Science Quarterly, 18, 75- 82.
  • Giorgi, A. P. (2000). Concerning the application of phenomenology to caring research. Scandinavian Journal of Caring Science, 14, 11–15.
  • Malterud, K. (2012). Systematic text condensation: a strategy for qualitative analysis. Scandinavian Journal of Public Health, 40, 795-805.
  • Phillips-Pula, L., Strunk, J., & Pickler, R.H. (2011). Understanding phenomenological approaches to data analysis. Journal of Pediatric Health Care, 25, 67-71. doi: 10.1016/j.pedhc.2010.09.004.

However, we acknowledge the need to provide improved clarity to the analysis section in order to offer a better description of the analysis process.

We have included this text in the analysis section:

Systematic text condensation (STC) is an elaboration of Giorgi’s principles (a follower of Husserl) [18], including four comparable steps of analysis [22]. In this descriptive approach, presenting the experience of the participants as expressed by themselves, and following Giorgi, STC provides an explorative proposal to present vital examples from peoples’ life worlds [22]. Also, like Giorgi’s method [19,20,23,24], STC implies analytic reduction with decontextualization and recontextualization of data [22]. This procedure consists of the following steps [22]: 1) reading all the material and an overview of the data; 2) reviewing the transcript line by line to identify meaning units. A meaning unit is a text fragment containing some information about the research question. Subsequently, coding begins (decontextualization), which includes identifying, classifying, and sorting meaning units and marking these with a code –a label that connects related meaning units into a code group; 3) implying the systematic abstraction of meaning units within each of the code groups established in the second step of analysis. Empirical data are reduced to a decontextualized selection of meaning units sorted as thematic code groups across individual participants; 4) data are reconceptualized, putting the pieces together again. Themes are developed, providing stories that reflect the participants’ experiences [22]).

Also, we have included new references:

  1. Phillips-Pula, L.; Strunk, J.; Pickler, R.H. Understanding phenomenological approaches to data analysis. J Pediatr Health Care 2011;25: 67-71. doi: 10.1016/j.pedhc.2010.09.004.
  2. Dowling, M. From Husserl to van Manen. A review of different phenomenological approaches. Int J Nurs Stud. 2007;44(1):131-42. doi: 10.1016/j.ijnurstu.2005.11.026.
  3. Norlyk, A.; Harder, I. What makes a phenomenological study phenomenological? An analysis of peer-reviewed empirical nursing studies. Qual Health Res. 2010;20(3):420-31. doi: 10.1177/1049732309357435.
  4. Malterud, K. Systematic text condensation: a strategy for qualitative analysis. Scand J Public Health 2012, 40,795-805. doi: 10.1177 / 1403494812465030.
  5. Giorgi, A. (2005). The phenomenological movement and research in the human sciences. Nurs Sci Q. 2005;18(1):75-82. doi: 10.1177/0894318404272112.
  6. Giorgi, A. P. (2000). Concerning the application of phenomenology to caring research. Scand J Caring Sci. 2000;14(1):11-5. doi: 10.1111/j.1471-6712.2000.tb00555.x.
  7. Results. Depending on the type of phenomenological approach used, I would expect more depth to interpretations of what the data/quotes meant.

Response: The reviewer is right. In the case of interpretive phenomenology, the results tend to be based on the process of integration and interpretation by the researchers, who make use of the participants' narratives. However, in the case of descriptive phenomenology, the interpretative exercise plays a secondary role and the codification process is more relevant by identifying meaning units, codes, and code groups. Subsequently, it is important that the data are reconceptualized, putting the pieces together again (synthesizing). By synthesizing the contents of the condensates, descriptions and concepts (Themes) are developed, providing stories that reflect the participants’ experiences (Malterud, 2012). There are more description of findings based on participants´ narratives that interpretation.

In the previous two questions, the coherence between the design, the type of analysis chosen, and the results has been explained.

Reference:

Malterud, K. (2012). Systematic text condensation: a strategy for qualitative analysis. Scandinavian Journal of Public Health, 40, 795-805.

  1. Discussion. I do not fully understand the origin of this assertion" Our results reported consideration of NTC as Advanced Practice Nursing (APN)". I believe that it is an interpretation of the researchers without a basis in the results and that it would be interesting to study it more deeply in future research.

Response: We have modified this paragraph.

We have included the following text:

The authors of the present study believe that NTCs should be considered as Advanced Practice Nursing (APN). Thus, APN is described by two key factors: the expansion of competences beyond the usual range and an orientation towards the promotion of the profession and its holistic values, focused on health and caring for the person [28]. The Spanish consensus of competencies for APN is based on 12 domains (54 competencies) which include: evidence-based research and practice, clinical leadership and consulting, quality management and clinical safety, and autonomy for professional practice, among others [29,30].

5.With regard to the limitations of the study, I do not believe that the generalization of the results to other populations is a limitation, because the main characteristic of qualitative research is the study of the local and specific.

Response: We agree with the reviewer. We have removed the following sentence: “This study has some limitations concerning generalizability that limit the extrapolation of our results to the whole population.”.

We hope that you are satisfied with this revision and that the manuscript is now suitable for publication in IJERPH.

Sincerely,

The Authors
